# Perceptions and attitudes towards unmanned aerial vehicles (drones) use for delivery of HIV medication among fisher folk communities on the Islands of Kalangala, Uganda

Jackie Lydia N. Ssemata[1], Rachel King[1,2], Patrick Ssesaazi[1], Agnes Bwanika Naggirinya[1], Joshua Beinomugisha[1], Rosalind Parkes-Ratanshi[1,3]*

1 Academy for Health Innovation, Infectious Diseases Institute, Makerere University Kampala, Kampala, Uganda, 2 Department of Epidemiology and Biostatistics, Institute for Global Health Sciences, University of California, San Francisco, California, United States of America, 3 Department of Psychiatry, Institute of Public Health, University of Cambridge, Cambridge, United Kingdom

* rp549@medschl.cam.ac.uk/rratanshi@idi.co.ug

**Data Availability Statement:** Anonymized data will be made available to interested researchers

## Abstract

The study aimed to assess the attitudes of stakeholders towards the use of unmanned aerial vehicles (medical drones) for delivering antiretroviral therapy (ART) in the Kalangala district of Uganda, which is comprised of 84 islands and has approximately 18,500 People Living with HIV (PLHIV). A qualitative baseline study was conducted to assess the acceptability and feasibility of using a medical drone for ART delivery in the island settlements of Kalangala Islands. The data revealed four emerging themes: knowledge about the drones, perceived benefits of medical drone delivery, perceived risks of medical drone use, and recommendations for future use. The study found that most participants, especially healthcare workers and key opinion leaders, were aware of the medical drones, which could reduce transport costs, deliver medication on time, and reduce healthcare workers' workload. However, there were also perceived risks related to the use of medical drones, such as stigma, reduced contact with healthcare providers, and maintenance and security issues. The study provided evidence that medical drones would be acceptable and have support from various stakeholders in the island settlements for ART delivery. However, concerns were raised about potential stigma and less health worker interaction. This qualitative work allowed the team to address these concerns during the pilot phase.

## Background

The Joint United Nations Programme on HIV/ AIDS (UNAIDS) strategy for elimination of HIV as a global public health threat is to achieve 95% of People Living with HIV (PLHIV) knowing their HIV status, 95% on treatment and 95% achieving viral suppression (95:95:95 targets) [1,2]. According to Uganda Population-based HIV Impact Assessment (UPHIA) 2020, the overall prevalence of HIV/AIDS in the general population was estimated at 5.5%. For

through request to the PI and completion of a data sharing agreement between institutions as per Uganda National Council of Science and Technology and Uganda Data Protection Act. Data can be accessed on 10.5061/dryad.fqz612jzf Data can be accessed through the Infectious Diseases Institute Research office from Paul Gonza - pgonza@idi.co.ug and Dr Agnes Bwanika Naggirinya -anaggirinya@idi.co.ug.

**Funding:** $1,100,000 from Janssen Global Public Health to PRR. Funders advised on the implementation of the drones project, but not on the research aspects. They were not involved in data collection, analysis or publication review. The funders had no role in study design, data collection and analysis, decision to publish, or preparation of the manuscript.

**Competing interests:** I have read the journal's policy and the authors of this manuscript have the following competing interests: Rosalind Parkes-Ratanshi has received grant funding from Janssen GPH and from Pfizer Pharmaceuticals. All other authors have declared that no competing interests exist.

adults (15 years and above), the prevalence was estimated at 5.8% with higher prevalence in women (at 7.2%) than in men (4.3%). The adult Viral Load Suppression (VLS) was 75.4% which is below 95% target [3]. In order to achieve the UNAIDs target, PLHIV need to be engaged throughout the HIV care continuum and have life-long retention in care.

Client retention in care is critical to improve adherence levels which is the cornerstone for achieving targeted viral suppression rates. Recent reports (2018) indicated low retention rates for clients initiated in HIV care, ranging from 65–75% against the UNAIDS 95% target [4,5]. Retention in HIV care and adherence to HIV medication is affected by both individual and health system level barriers including (but are not limited to) long distances with high cost of travel, long clinic waiting times, stigma and frequent ART stock-outs [5]

To address these barriers, the World Health Organization (WHO) and governments (including Uganda) have endorsed differentiated service delivery (DSD) models for HIV care, a patient centered ART delivery approach [6,7]. DSD models enable clients to access ART refills at flexible refill points within their communities thereby addressing barriers related to accessibility of care. The recommendations are aimed at decongesting facilities and ensuring efficient HIV services delivery, and supporting PLHIV to access ART close to their homes.

The challenges of achieving 95:95:95 are multiplied in geographically hard to reach areas such as islands and mountains, especially in low income countries due to ART delivery being challenging and expensive. This results in poor adherence and high patient attrition rates. Furthermore, DSD can be difficult to implement in remote and hard to reach areas as the density of PLHIV is lower, and each person is harder to reach. This leads to the DSD model delivery of refills not to be consistently achieved due to multiple challenges. These include individualized and community-level stigma, fear of detachment from health facilities insufficient training of health workers in DSD delivery, and insufficient funding to fully operationalize community drug pick-up points [8].

Kalangala district is a district consisting of 84 islands in Lake Victoria. It has approximately 18,500 PLHIV, with an adult HIV prevalence between 18–25%; compared to national prevalence of 5.5% [9]. HIV treatment outcomes in this region are also poor; according to program data, with 24% of the PLHIVs being lost to follow up from care by 6 months. While 78% of PLHIV in care have a suppressed viral load this is an overestimate, as only around 50% undertake viral load testing as planned, and receive their results [10].

Medical drones are being explored as possible solutions to the geographical barriers associated with delivery of medical supplies in areas with poor infrastructure or geographical limitations. We are undertaking a pilot project to see if medical drones can directly deliver HIV medication refills to patients at landing/other remote sites. The drone will deliver drugs to peer groups under community client led ART delivery (CCLAD). The ART clients will form groups of 7 people maximum, choose a group leader that will pick their drugs from the medical drone and take refills on their behalf at time and place of convenience. The medical drone will deliver at a delivery point in a settlement that will be mapped out.

Flight routes from the health center to the delivery points will be input into the medical drone system. The medical drone will detect a QR coded piece of cloth at a delivery point and will land at where the cloth is placed (precision landing). The possible benefits of this include, saving clients a trip to health facilities if an outreach does not happen. This may improve accessibility of ART with resultant improvement in patient retention in care, ART adherence and viral suppression rates. By reducing the time taken on ART refills at outreaches, the medical drone use may increase capacity of healthcare workers in remote areas to undertake other activities such as vaccinations or antenatal checks at outreaches. Studies have shown that a total of 2,066 drowning cases were reported with 1,332 (64%) deaths in 14 districts around lake Victoria [11,12]. Medical drone ART delivery may reduce the risk of drowning and

accidents with boat travel. However, in delivering directly to patients, there are possible concerns about privacy, security, safety, and the long-term sustainability of the drones for health projects [13].

Despite the growing interest in medical drones, there is limited data on perceptions of patients on use of drones. To inform the pilot project we conducted a qualitative sub-study in Bufumira Sub-County within Kalangala District involving people living with HIV and those not living with HIV. This study aimed at exploring the perceptions and attitudes towards the use of medical drones for ART delivery in on the Islands of Kalangala, Uganda.

## Methods

### Study design

This was an exploratory, cross-sectional, qualitative study using FGDs and IDIs as data collection methods. This was carried out as part of a three-stage study entitled Overcoming Geography with Technology, which included phase one baseline data collection, phase two pilot drone deliveries, and phase three cluster randomized trial. This baseline sub-study was situated in phase one (pre-drone deployment stage) designed to explore perceptions and attitudes towards the use of medical drones for ART delivery in on the Islands of Kalangala, Uganda.

### Study setting

The interviews were conducted with participants from Bufumira subcounty in Kalangala district. Bufumira Government Health Centre III was the planned site of the medical drone pilot study, and the health centre serves eight settlements on surrounding islands. An island settlement is an area of an island including a strip of beach where boats dock, and where the population on the island tends to live; there may be more than one island settlement on any island. This study was conducted in the catchment area of a rural health center facility (Bufumira HCIII) at the island settlements of Kitobo, Kusu, Kaazi, Mukaka, Ssanyu, Bbanda and Bufumira "Fig 1,"

### Participants and recruitment

A total of 76 participants were purposively selected for FGDs from seven island settlements. Seven FGDs (n = 60) were conducted including; pregnant women living with HIV, pregnant women not living with HIV at Bufumira, health care providers at Bufumira, youth living with HIV and youth not living with HIV at Bufumira. Key opinion leaders and PLHIV at other island settlements getting care from Bufumira were also included.

To recruit participants (People living and not living with HIV), information about the study was provided to people attending Bufurima health facility as a group information session. Kitobo and Kaazi island settlements had the largest population of PLHIV, so we recruited directly from these sites for adult PLHIV groups.

These were then invited to share their perceived thoughts and opinions about using medical drones to deliver ART. To ensure representative data, the social science team (JS, JB, ON) purposively recruited participants according to age, sex and location.

Twelve IDIs with key opinion leaders, including Local Council chairmen, police officers, religious leaders, local administration officers and Health Care workers were conducted (Table 1). These were purposively selected by the social scientists together with the local leaders according to their influence to the general public and had to have rich information about the population due to their role in the community. They were found by virtue of their influence. Others were listed by some residents as people they listen to and who would well

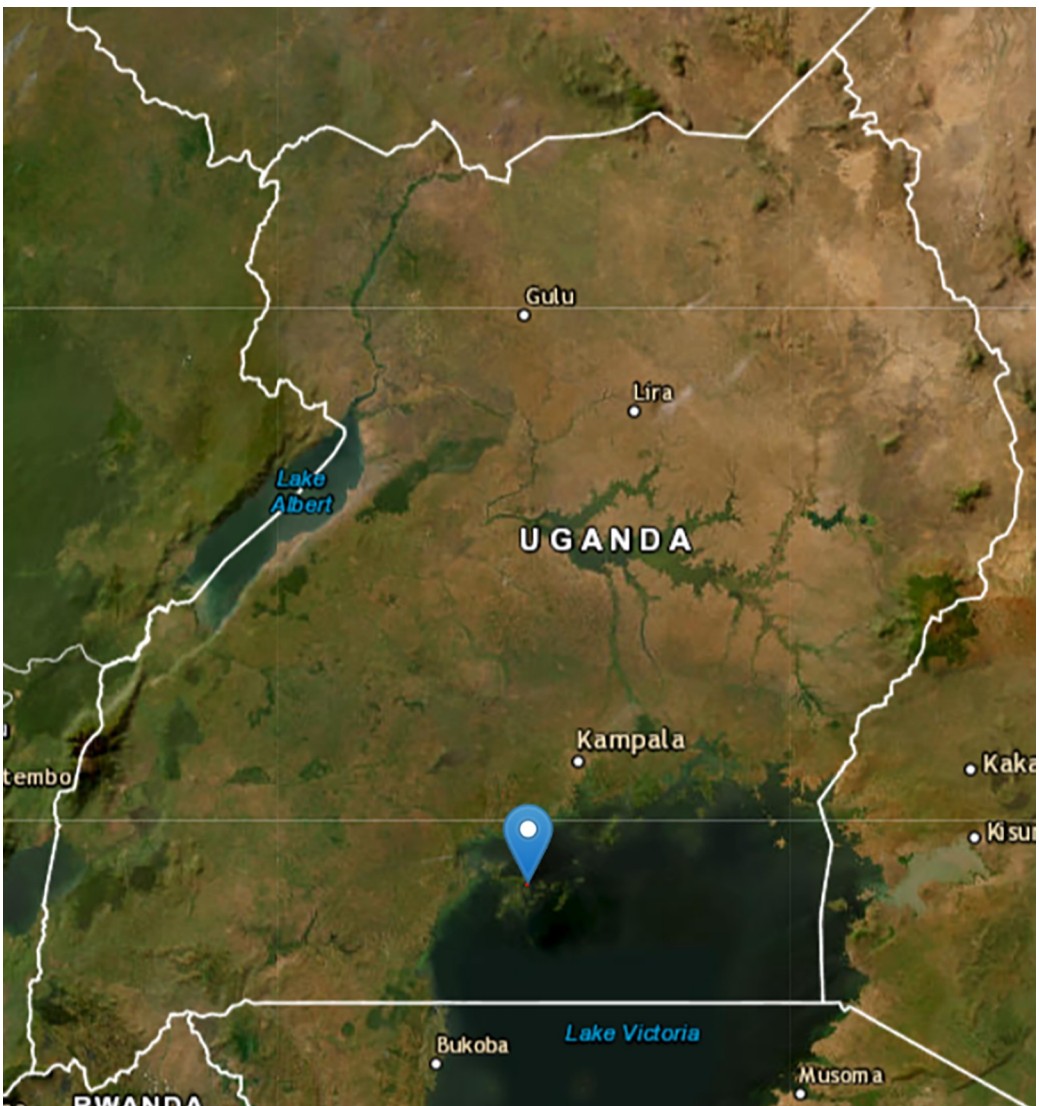

**Fig 1. Map showing the location of Bufumira in Uganda.** Source: https://earthexplorer.usgs.gov/.

represent their ideas. The key opinion leaders were spread across the subcounty on the island settlements of Bufumira, Kaazi, Kuzu, Sanyu, Bbanda and Mukaka and therefore we undertook in depth interviews with these.

In Kitobo however, there was a large number of opinion leaders and so an FGD was also undertaken for them. We recruited participants for FGD and IDI until saturation was reached.

Participants that showed interest in both the FGDs and IDIs were screened as per inclusion and exclusion criteria before written informed consent was obtained. Those eligible had to be 18 years and above or emancipated minors (15 years to 18 years) who were receiving ART from Bufumira health centre III (HCIII) or were not living with HIV but received care from the Bufumira HCIII. They should have been a health care worker at Bufumiira HCIII, should have stayed in Bufumira sub county for at least 12 months and are willing to stay for another 12 months, were considered a key opinion leader and should have provided a signed consent form. Participants were excluded if they did not meet the above criteria, give consent or were known by the community as having a mental illness.

**Table 1. Participant characteristics in focus group discussions and in-depth interviews, September-November 2020, Kalangala, Uganda.**

| FGD | Age range | No. of participants | Male | Female | Status/ Composition | Location |
|-----|-----------|---------------------|------|--------|---------------------|----------|
| 1 | 20–38 | 10 | 0 | 10 | Pregnant Women Living with HIV and those not living with HIV | Bufumira |
| 2 | 18–47 | 12 | 5 | 7 | Healthcare workers | Bufumira |
| 3 | 18–23 | 6 | 2 | 4 | Youth Not living with HIV | Bufumira |
| 4 | 18–33 | 6 | 3 | 3 | Youth living with HIV | Bufumira |
| 5 | 32–67 | 6 | 3 | 3 | People Living with HIV | Kaazi |
| 6 | 28–58 | 10 | 7 | 3 | Key Opinion leaders | Kitobo |
| 7 | 33–49 | 10 | 3 | 7 | PLHIV | Kitobo |
| **Total** | | **60** | **23** | **37** | | |
| IDI | 32–40 | 6 | 5 | 1 | Key Opinion leaders | Bufimira, Mukaka, Sanyu |
| | 41–50 | 2 | 2 | 0 | Key Opinion leaders | Kitobo, Kusu |
| | 51–60 | 1 | 1 | 0 | Key Opinion leader | Bufumira |
| | 61–70 | 2 | 2 | 0 | Key Opinion leaders | Bufumira, Kusu |
| | 71–75 | 1 | 1 | 0 | Key Opinion leader | Bufumira |
| **Total** | | **12** | **11** | **1** | | |

## Data collection procedures FGD notes/moderate, IDI one to one

Data collection took place between 1st September 2020 and 30th November 2020. The interviews were conducted by two trained social scientists (JS, JB) with experience in qualitative research. The interviewer started the interview by introducing a general conversation to create rapport. Participants were assured of anonymity. All interviews were conducted face to face, in both English and Luganda languages using a semi-structured interview guide that was developed by the lead social scientist (JS).

The interview guide was developed with consideration of concerns and guide from the community leaders that were engaged at the study initiation phase.

We included open-ended questions like what they thought drones were, what they are used for, if they have ever seen them before, how they feel about the drones delivering their ART and what they recommend the means of delivery would be.

Data collection was conducted in quiet and convenient spaces to ensure privacy with the FGDs lasting between 60–120 minutes and in-depth interview sessions lasting between 45–60 minutes. All IDIs and FGDs were audio-recorded with permission from the participants. For FGDs the social scientists worked together, with one asking questions and the other taking notes. For IDI these were done with interviewees on a one to one basis to increase rapport.

Interviewers asked participants a series of general questions about the drones from the interview guide. We later explained the intended use of the drones: participants that the drones were going to deliver ART from the Heath Center to the island settlements and asked them about the perceived benefits and risks of using drones for ART delivery and how else the drone could be of help in their community. At the end of the interview, all participants were reimbursed for their time and transport a sum of 20,000/ = (approximately $5.5).

## Data analysis

Prior to data collection, the study team including two graduate social scientists were trained on the study objectives, the consenting process, confidentiality of participant and respondent data, data collection tools and the procedures for administering the tools. FGD and IDI recordings were transcribed verbatim and those conducted in Luganda were later translated into English. Data was cross-checked and verified by lead social scientist (JS) for consistency and accuracy.

Preliminary data analysis occurred concurrently with data collection. The lead social scientist (JS) started the process of typing and reading through the transcripts and notes. The second social scientist (JB) reviewed the transcripts and developed the codebook.

The codes from the two social scientists were discussed in three research team meetings which helped draw attention to emerging themes. This was an important part of the preliminary analysis and identification of thematic categories. The list of themes was generated during the process providing a framework for a codebook. The data were processed using NVivo software version 12 (QSR International Pty Ltd). Thematic content analysis procedure was followed for the analyses of the data. Quotations illustrative of participant's experiences regarding the issues being studied was identified and used in the presentation of findings.

## Ethical considerations

Ethical approval was obtained from Makerere University School of Public Health (MUSPH) research and ethics committee (MakSPH-REC 801). Additional approvals were obtained from Uganda National Council for Science a Technology (HS725ES). Informed written consent was obtained from every participant prior to participation; and, the ability to withdraw from the study at any point was communicated. Participants were also assured of confidentiality and the anonymity of recordings; their participation and withdrawal were voluntary, and that gathered information would be used solely for the purpose of the study.

## Results

We report on the findings from 72 participants living on the seven Island settlements. Sixty (60) participants in total attended the FGDs as summarized in Table 1 below. Composition of the FGDs was mostly female (37). Twelve participants took part in the in-depth interviews, all but one was male. Saturation was reached after the interview of the 12th participant for the in-depth interviews.

1. **Knowledge and familiarity of medical drones**

Across all island settlements, awareness and knowledge about drones was similar. Healthcare workers and key opinion leaders were more aware of medical drones than other participants. Most of the community members, across sites, had heard about drones from films and seen them in action at functions taking photos or recording videos.

*"We were in Kalangala for a function. That is when I saw it taking videos and photos and it was being controlled by someone"* **PLHIV FGD, Bufumira**

*"What I know about medical drones, is that it is going to be here and will be used to deliver medicine to the islands where masts have been constructed."* **(IDI 03 Male)**

The varied responses may be attributed to the fact that the healthcare providers and key opinion leaders had been sensitized/ oriented about the medical drones and what they would be used for. The sensitization started in January 2020 and was continuous process. The sensitized started at the time the weather masts were being erected and permission was sought from the leaders. They were told about the possibility of bringing medical drones to the islands to deliver drugs.

2. **Perceived benefits of drone delivery of ART**

After explaining about medical drones to the participants, they were generally supportive of them and mentioned some perceived benefits.

### 2.1 Efficiency during bad weather

Bad weather especially the rainy season, being one of their main challenges to receiving ART, few of the participants perceived the medical drones as an aircraft that could not be affected by weather. Most were sure to receive their medicines despite the weather. *"I would prefer the drone delivering ART because if for example the health workers were supposed to come and it begins raining and stops like at 4 PM, the health workers will fail to come but the drone certainly will deliver the medicine"* **PLHIV FGD, Kitobo**

One of the participants though mentioned that medical drones are also affected by weather and could not deliver ART in such situations.

### 2.2 Reduced transport costs

Most of the health providers believed that the medical drones would reduce the costs of transport in terms of number of trips and would help them pick up supplies if they were missing medication.

> *"No cost, I don't think it will charge anything, yet it will deliver medicine where I stay without me injecting anything. I would save that money and it boosts me in something else."* **(IDI 06 Female)**

For the opinion leaders, their interest was in the medical drone being cost effective. Many mentioned that boat transport is very expensive and includes boat hire, paying the coxswain and buying fuel. Even with the taxi boat, they still have to collect money as a group to pay for the above at a reduced price but still expensive. With the medical drone in place, they believed that it would help reduce the costs as transport to the health center would be reduced.

> *"So, I think the drone will be very beneficial, cost-effective, and fast, and like you know time is money."* **Key opinion leaders FGD, Kitobo**

### 2.3. Safety

Most of the participants mentioned that the drone would be fast compared to the boat in delivering ART and their drugs would be safe. They were convinced that the drone would carry medicine that is enough for everyone.

> *"These drones of ours will be very beneficial because the medicine will be delivered when it is safe. Even if we are saying that its carrying capacity is small but I know it will deliver the medicine which is enough for everyone."* **PnotLHIV Youths FGD, Bufumira**

### 2.4 Timely delivery of medication

All perceived the drone as being fast and therefore they would receive their medication on time despite the weather as long as they communicate their need at the time. Furthermore, ART reaching on time would allow people to go and do their other work instead of waiting for deliveries to arrive.

> *"I am saying the medicine will be delivered on time. It will just be a matter of calling and requesting for medicine and it is delivered on time. Whether it is raining, or whether there is no fuel or boat, medicine will be delivered on time"* **PnotLHIV Youths FGD, Bufumira**

### 2.5 Improved adherence through improved access to medication

Almost all participants understood that if the drone was to deliver their medicines, then a peer would receive on their behalf and then distribute it. They believed that through drone and peer delivery, they would miss fewer appointments and thus take their ART more regularly.

*"Yes, it is going to help us not to miss appointments because even if you are not around, the drone brings your medicine and the peer supporter collects. The peer supporter then brings it over to my home."* **(IDI 03 Male)**

### 2.6. Reduced HCW workload

Using the drone to deliver ART was seen as a mechanism to reduce the workload of the HCWs. All the HCWs believed that with the drone, outreaches could focus on other services rather than ART refills which currently takes up most of their time. They reported that when they go for outreaches there is barely any HCW left at the Health Center to serve the remaining population. HCWs perceived that the drone would reduce the hospital lines and the number of people at the Health Center thus reducing their workload.

*"It will be so good because it will reduce the time we will spend working on these many clients because the medicine can be sent for the stable clients. The team will now only work on those who are non-suppressed and who are just a few clients. This way, time will even be available for us to adequately interact with these clients as opposed to working on very long lines. It will also reduce the costs of going to Kalangala for some deliveries because it will be doing the picking and delivering"*

*(HCWs FGD, Bufumira)*

### 2.7. Delivery of other medicines

Most were happy about the drones delivering medicines and thought that they may be able to transport other drugs apart from ART. They believed that with the drone in place they could now receive drugs that they could not access due to transport challenges.

*"To add to that, since the drone is going to be delivering both ART and other medicines, I as a patient with a kidney problem will be happy because it will be delivering that medicine and I will now be in a position to access it."* **PLHIV Youths FGD, Bufumira**

Almost all participants perceived that the drone could help in emergencies and carry other medicines that are needed through a phone call. The HCWs mentioned that they could use the drone to carry the medicine or documents that they would have forgotten at the Health center as they travelled to different island settlements.

### 3. Perceived risks of drone delivering ART

Most of the participants agreed that the medical drone project could pose some risks. The perception of risks varied between different study participants and included issues around stigma, reduced contact with health care providers and thus a reduction in quality of care, maintenance and security of the drones themselves and potential conflict of interest from health care providers who may perceive their income could reduce if the drones provide the outreach they would have otherwise provided.

### 3.1 Stigma

Some of the participants showed concern if the medical drone was to deliver only ART citing stigma from their friends and family.

> *"Every time the drone comes they will be telling us; your drone has come, haven't you seen it? . . .everyone stigmatizing against us"* **PLHIV FGD, Bufumira**

The participants on ART worried that if the drone carried no other medical supplies then they would associate the drone with those on ART and yet if it carried other supplies then they could not associate it to HIV/ART.

### 3.2. Reduced contact with Health care workers

Many believed that the drone would reduce the contact between the HCW and the Client as they would just deliver drugs. This would mean that those using the drone would take longer to meet with the HCWs at health facilities.

> *"The first risk is clients being cut off from their health workers, the drone only takes medicine and returns, doesn't carry a person yet the clients need their health workers"* **PLHIV FGD, Bufumira**

A few participants showed concern regarding contact with the HCWs. They wondered that if the drones were going to deliver ART, when would they meet the HCWs? A number of them said that it was one main reason that they would not want the drones to deliver ART. Some mentioned that they could have other ailments that they would want to share with the HCW but will not be possible as the drone will not be moving with a HCW.

### 3.3. Security and maintenance of the drone

Some worried that the drone could be stolen and suggested hiring a security guard to watch over it.

> *"We don't have proper security for it. It would need a security guard to protect it"* **HCWs FGD, Bufumira**

Few were concerned about the drones' maintenance and what could happen in case it got any technical issues.

> *". . .in case of a breakdown while in transit (in the air), and it malfunctions just like you know systems get breakdowns. . .all that was being delivered is lost"* **HCWs FGD, Bufumira**

### 3.4 Sabotage by Healthcare workers

Some key opinion leaders spoke about sabotage of the medical drone by the health care workers. As health care providers receive an allowance every time they go for an outreach, with the drone in place, the number of outreaches could be reduced which could reduce the allowance that they receive.

*"The health workers themselves may sabotage it because they have been getting some allowance when they come for an outreach activity and that will stop. . ."* **Key opinion leaders FGD, Kitobo**

Some of the participants especially from the FGDs believed that would be hard to maintain the drones in case of an accident or service as there is no one on the island with such expertise. Medicine would all be lost in case of an accident as it would all be destroyed.

4. **Recommendations**

We asked for recommendations about what services the medical drones could provide. As medical drone drug delivery is such a new and perceived highly technical project, only a few recommendations were suggested.

### 4.1 Transporting other medical supplies

To avoid stigma, many recommended that the drone carries other medical supplies.

*". . ..it can bring other medicines like that of malaria. Since there are village health teams (VHTs) in the villages, they can receive it and dispense to those in need. Panadol, Coartem and other simple drugs can be delivered by the drone. . .. . ."* **(IDI 08 Male)**

Some of the participants were unaware of what the medical drones could carry. They mentioned that the drones could help transport patient documents and from one FGD a respondent thought that it could help carry a pregnant woman to hospital.

They recommended sensitization, working hand in hand with the VHTs and making sure that people understand what exactly the medical drone would be delivering.

Main themes identified from both the FGDs and IDIs are summarized in Table 2 below.

## Discussion

Our findings highlight the perceived benefits and risks of medical drone use among a variety of participants in a hard-to-reach region of Uganda. We found evidence of perceived benefits around many critical elements in health service delivery including, reduced workload for health care providers, timely access to medication, reduced cost in transport, improved adherence to medical appointments. These are key factors that address important health system barriers in

**Table 2. Summary of main themes and sub-themes.**

| Theme | Sub-Themes |
|---|---|
| **Knowledge about the drones** | •Seen in Films<br>• Takes photos and videos<br>• For surveying land |
| **Perceived Benefits of medical drones Delivery** | •Efficiency during bad weather<br>•Reduced transport costs<br>•Increased safety<br>•Timely delivery<br>•Reduced missed appointments<br>•Reduced workload for HCWs<br>•Delivery of other medicines |
| **Perceived risks of medical drones Delivery** | •Reduced contact with HCWs<br>•Security of the medical drone<br>• Stigma |
| **Recommendations for future Drone use** | Transporting other medical supplies |

this region of the country. The perceived risks that of potential for reduced contact with health workers if medical drones efficiently deliver ART, possible increased stigma if medical drones only deliver HIV drugs and a fear of keeping the medical drone safe were mentioned. These perceived risks are understandable and important but are not complicated to mitigate.

Medical drone use could become one of the most effective and efficient answers to widespread health coverage, where terrain, distance and mobility are key obstacles to meeting healthcare targets. [14,15] In Rwanda medical drones are already delivering 60% of national blood supply outside the capital, Kigali, and approximately half being used in cases of postpartum hemorrhage. The technology has allowed the blood delivery time to be cut from 4 hours to 15 minutes. [16] Drones have also been used in delivery of anti-venom in instances of snake bites, delivery of vaccines and infection and control measures during COVID-19. [16,17].

A recently published study from Germany undertook a qualitative analysis of patient perceptions for medical drones in COVID -19 which was overwhelmingly supportive, but highlighted many gaps needed for full implementation [18]. Others have looked at health worker perspectives [19]. Our study is the first exploring perceptions and acceptability of medical drones for support of HIV care delivery, which is the largest cost driver of health care budgets in Africa. Our aim was to use results of the study to inform a pilot of medical drone use for HIV medication (ART) delivery. Our findings have highlighted more perceived benefits than risks and challenges in drug delivery through the use of medical drones. Community and health provider mentioned willingness to use medical drones for a few clear and logical reasons. The experience of inconsistent access treatment has led to appreciation for innovative alternatives to safe drug delivery options. Participants believed that drone drug delivery would be safer (for the drugs and people), more efficient, more cost-effective, and less taxing on health workers all leading to an improvement in overall drug adherence.

There were, however, some unique challenges that our participants noted. One concern was that people living with HIV do value face to face interaction with health care providers and use of drone drug delivery would potentially limit the necessity of seeing health workers regularly. Others studies noted challenges such as infrastructural, financial, technological barriers, security issues with respect to both participants and the drone itself, limited or lack of policies, guidelines and frameworks in Sub-Saharan Africa governing the ethical and effective use of drones in healthcare. [20]

Smith et al have highlighted that public acceptance is important in order that we move the debate on drones for public good forward. We believe that this type of engagement and documentation of concerns of beneficiaries of medical drones through qualitative research is essential to enable community voices to be heard in program design. The study provided important information on concerns raised that needed to be carefully addressed by the medical drones' project team. This included possible stigmatization of PLHIV through use of the drones. To counter this in our drone pilot we recruited those peer support workers who were open about their HIV status and had already disclosed their status publicly to be the people receiving the drones. Secondly there was concern about loss of outreach activities which could lead to less health worker interaction with patients and also loss of income for HCW. This led to our pilot project evaluating how the medical drones could support the existing outreach activities, by reducing the time needed for ART refills during outreaches, and freeing up time for other activities such as antenatal checkups, childhood vaccinations, screening for non-communicable disease and more.

## Limitations of the study

Prior to the study we had already started sensitization of health care workers and the community leaders on the proposed project, and so this may have influenced the responses from

health care workers. This was during the initiation stage as we needed to get their permission, support and engagement which is very helpful. The information is likely to have filtered down to some PLHIV and other users of the health facility as the project was causing some excitement. As external health care workers, the team are seen as related to the government, and so some issues may not have been expressed for fear of repercussions. For example, in community meetings fishermen expressed that the drones may be used for surveillance for illegal fishing, yet this was not raised as an issue in the FGDs.

## Conclusion

This study provided reassurance that the medical drones project would be well received by intended beneficiaries and that the team would have support in the community. This sub study provided very valuable information to support project implementation, and we recommend others planning medical drone projects to undergo similar qualitative work to understand the benefits and risks that this novel technology can bring to hard to reach populations. One of the valuable information that we got was to use the drones to deliver other medical supplies like blood so that the people receiving drones are not stigmatized.

## Supporting information

**S1 Text. Interview guide.**
(PDF)

## Acknowledgments

We would like to acknowledge study participants, Kalangala district leadership, funders and study staff -Joan Akullo, Dickson Masoni and Paula Muhawe.

## Author Contributions

**Conceptualization:** Rachel King, Rosalind Parkes-Ratanshi.

**Funding acquisition:** Rosalind Parkes-Ratanshi.

**Investigation:** Rosalind Parkes-Ratanshi.

**Supervision:** Patrick Ssesaazi, Rosalind Parkes-Ratanshi.

**Validation:** Agnes Bwanika Naggirinya.

**Writing – original draft:** Rachel King, Agnes Bwanika Naggirinya, Rosalind Parkes-Ratanshi.

**Writing – review & editing:** Jackie Lydia N. Ssemata, Patrick Ssesaazi, Agnes Bwanika Naggirinya, Joshua Beinomugisha, Rosalind Parkes-Ratanshi.

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
