## [Decision Letter · Decision Letter 0]

6 Dec 2023

PGPH-D-23-01417

Perceptions and attitudes towards unmanned aerial vehicles (drones) use for delivery of HIV medication among fisher folk communities on the Islands of Kalangala, Uganda

Dear Dr. Parkes-Ratanshi,

Thank you for submitting your manuscript to PLOS Global Public Health. After careful consideration, we feel that it has merit but does not fully meet PLOS Global Public Health’s publication criteria as it currently stands. Therefore, we invite you to submit a revised version of the manuscript that addresses the points raised during the review process.

We look forward to receiving your revised manuscript.

Kind regards,

Sanghyuk S Shin

Academic Editor

Journal Requirements:

1. Please send a completed 'Competing Interests' statement, including any COIs declared by your co-authors. If you have no competing interests to declare, please state "The authors have declared that no competing interests exist". Otherwise please declare all competing interests beginning with twhe statement "I have read the journal's policy and the authors of this manuscript have the following competing interests:"

b. If any authors received a salary from any of your funders, please state which authors and which funders.

3. Please provide separate figure files in .tif or .eps format only and remove any figures embedded in your manuscript file. Please also ensure all files are under our size limit of 10MB.

Additional Editor Comments (if provided):

Please review carefully each of the comments and suggestions from the reviewers. In addition, please ensure that the reporting is consistent with best practices in reporting qualitative research as found here: https://www.equator-network.org/reporting-guidelines/coreq/.

Reviewers' comments:

Reviewer's Responses to Questions

**Comments to the Author**

1. Does this manuscript meet PLOS Global Public Health’s publication criteria? Is the manuscript technically sound, and do the data support the conclusions? The manuscript must describe methodologically and ethically rigorous research with conclusions that are appropriately drawn based on the data presented.

Reviewer #1: No

Reviewer #2: No

Reviewer #3: Yes

Reviewer #4: Yes

2. Has the statistical analysis been performed appropriately and rigorously?

Reviewer #1: N/A

Reviewer #2: No

Reviewer #3: N/A

Reviewer #4: N/A

3. Have the authors made all data underlying the findings in their manuscript fully available (please refer to the Data Availability Statement at the start of the manuscript PDF file)?

Reviewer #1: No

Reviewer #2: No

Reviewer #3: Yes

Reviewer #4: Yes

4. Is the manuscript presented in an intelligible fashion and written in standard English?

Reviewer #1: No

Reviewer #2: Yes

Reviewer #3: No

Reviewer #4: Yes

5. Review Comments to the Author

Reviewer #1: Thank you for allowing me the opportunity to review this interesting manuscript, Perceptions and attitudes towards unmanned aerial vehicles (drones) use for delivery of HIV medication among fisher folk communities on the Islands of Kalangala, Uganda. The topic is timely and of significant societal importance. Adherence to HIV medications is at the backbone of ensuring the UNAIDS goals. Despite these strengths, there were a number of significant issues that I hope will help you strengthen this manuscript.

1) This paper will require a thorough editorial review in terms of grammar, spelling, double wording, and sentence structure in places. Please make sure all acronyms are spelled out (i.e., UPHIA, page 6, WHO, etc),

2) On top of page 6, there appears to be HIV cited, but with differing numbers.

3) Page 6, second para, which aim?

4) In the literature review, there is an entire section on barriers to HIV adherence which appears to be missing. This will be foundational to your study.

5) HIV adherence data is not presented in the background as well.

6) Last para on page 6, I believe there is inconsistent information on prevalence of HIV again.

7) No information on prevalence of drowning events mentioned on page 7

8) Under Methods, what is the type of qualitative approach used. There are different types which have nuanced steps in completing.

9) Page 8, how were participants selected? What approaches were used?

10) Page 9, any information about the sites in terms of demographics, prevalence of HIV, etc

11) What were inclusion and exclusion criteria?

12) Page 9, first full para, last sentence. There is a lack of clarity on how participants were channeled into either focus groups or individual interviews. Now clear who participated in which approach.

13) Page 10, how was mental illness assessed and considered as an exclusionary criteria? Who recruited the participants?

14) Page 10, end of first para. Unclear how the semi structured interview guide was assembled and by whom? Were there suggestions by the community for types of questions to be asked?

15) Page 11, end of first two lines. Any processes for assuring trustworthiness and credibility of the data?

16) Page 12, Any data on sensitization as this may impact findings of the study as stated.

17) Page 18, A bit more information in the introduction on existing use of drones and how they are received by the community.

Reviewer #2: The paper has major methodological gap and lacks scientific rigor. The discussions and conclusion are not consistent with the results. The analysis has some deficiencies which in my view could be resolved by the authors. The study is a qualitative one and may only require including the tools and indicating availability of data upon request. This has not been made clear though. The paper poorly written grammatically. Require major English language editing.

Reviewer #3: The authors present their research on a novel and promising approach to delivering HIV medications to people living with HIV in islands that are difficult to reach by conventional transportation methods. The research supports the need to study medical drones and provides valuable insight into the acceptability of their use in the Islands of Kalangala. Despite some grammatical and punctuation issues, the paper is largely clearly written and well-organized, with findings that are highly relevant to ongoing challenges in HIV medication delivery.

General comments: Line numbers would be helpful for reviewing the manuscript. While errors were not substantial and did not get in the way of writers’ clear communication, the manuscript could benefit from some additional copyediting to address grammatical and punctuation issues.

Page 1, Abstract Introduction: Please define PLHIV where it first appears.

Page 1, Abstract Methods: Please define FGD where it first appears.

Page 3, paragraph 1: Please define UPHIA where it first appears.

Page 3, paragraph 1: By the following sentences: “For adults above 15 years the prevalence was 5.8%. HIV prevalence was estimated at 5.8% with the highest being in women at 7.2% than in men of 4.3%.” do the authors mean, “For adults above 15 years, HIV prevalence was estimated at 5.8% with higher prevalence in women (at 7.2%) than in men (4.3%).”?

Page 3, paragraph 1: What is the “90% target”? Is this different from the 95% targets mentioned in the paragraph before?

Page 5, paragraph 1: Should “Bufumira HCII” be “Bufumira HCIII” instead?

Page 6, paragraph 2: If possible, more information about how participants were assigned to FGD vs IDI would be helpful.

Reviewer #4: Perceptions and attitudes towards unmanned aerial vehicles (drones) use for delivery of HIV medication among fisher folk communities on the Islands of Kalangala, Uganda

PGPH-D-23-01417

Overall

The paper addresses an important issue on stakeholder perceptions on use of medical drones to deliver ART in hard to reach islands of Kalangala District. Authors should read the paper and edit it. There are several missing words and other editorial issues that should be attended to.

Here are specific suggestions for consideration.

Abstract

1. Write in full some abbreviated words on first mention eg PLHIV, FGDs

2. Results be consistent on use of caps. For instance in some instances ‘ medical drones’ in others Medical Drones

3. Replace ‘respondents’ with participants, ‘majority’ with most….given the qualitative nature of the study.

Background

1. Authors provide a good background that links the study to the 95,95,95 global targets.

2. These sentences ‘According to UPHIA 2020, the estimated prevalence of people living with HIV/AIDS was 5.5%. For adults above 15 years the prevalence was 5.8%. HIV prevalence was estimated at 5.8%’ are confusing give them a second look and edit accordingly. The HIV prevalence of 5.5% Vs 5.8% are these from different sources?

3. Recent reports indicated low retention rates for clients initiated in HIV care, ranging from 65-75% against the 90% target. Indicate the year.

4. Edit sentence starting ‘Furthermore, DSD can difficult (be is missing), in is repeated. Next sentence ‘is’ seems misplaced.

5. Paragraph starting ‘ Kalangala…….18-25%; delete ‘which’

6. 24% of PLHIV being lost to follow-up add ref

7. Second last paragraph under background; It would be helpful to describe the level at which the drones are planned to deliver ARVs. Will the delivery point be HCs, group setting, peer or individual households? Also explain how the delivery points will be located by the drone.

8. Last sentence ‘delivery’ in on (edit).

Methods

1. 1st sentence qualitative study ‘was’ is misplaced.

2. Delete the objective from study design, its already mentioned under background.

3. This sub study was implemented carried out (edit)

4. Study setting: edit to read as: eight settlements and surrounding

5. Page 6; key opinion leaders were selected delete (s)

6. Page , under data collection; We later explained the intended use of the drones….provide the explanation that was given to participants.

7. Results: These are generally clearly presented

8. Theme 1: The community members across……(were these most? A few?). Across all results authors should show the pattern in participants views.

9. 2.5 edit (adherence is repeated). The quote is more on missing appointments than adherence.

10. Some descriptions such as Programmers’ officer, male (could be identified). Who are these program officers and how many are they? Also page 14, assistant fisheries Officer, male… could be identified. Add this description under methods to re-assure the readers that these officials cannot be identified.

11. 2.7 replace majority with most.

12. 3.3. The description is too short to stand on its own as a sub-section and paragraph. Consider expanding it or merging it with another section.

13. Table 2, should be brought at the start of results section, referred to in text and with few sentences highlighting key characteristics in the table.

14. Discussion: This section should be strengthened. Authors should start with a brief summary of key findings and what they mean before moving on to general literature on medical drones.

15. The 1st sentence, paragraph 2, starting ‘despite the growing interest…….fits in background.

16. Next sentence: A recently published study from Germany…..1st present your findings and what they mean then compare with other studies.

17. Page 16, edit the 1st paragraph. Acceptability of medical (drone is missing), …..which is largest (the is missing before largest). Our findings have highlighted …….then (should be than). Read and edit the paper.

18. As mentioned, there are several interesting findings which authors have not discussed.

19. Are there any strengths of this study? Authors provided an explanation about medical drones could this have biased stakeholders?

20. Conclusion: The study provided valuable information to support project implementation……Explain briefly here or under discussion what information you mean and how helpful it was during implementation.

6. PLOS authors have the option to publish the peer review history of their article (what does this mean?). If published, this will include your full peer review and any attached files.

**Do you want your identity to be public for this peer review?** For information about this choice, including consent withdrawal, please see our Privacy Policy.

Reviewer #1: No

Reviewer #2: **Yes: **Albert Apotele Nyaaba

Reviewer #3: No

Reviewer #4: No

---

## [Decision Letter · Decision Letter 1]

21 May 2024

PGPH-D-23-01417R1

Perceptions and attitudes towards unmanned aerial vehicles (drones) use for delivery of HIV medication among fisher folk communities on the Islands of Kalangala, Uganda

Dear Dr. Parkes-Ratanshi,

Thank you for submitting your manuscript to PLOS Global Public Health. After careful consideration, we feel that it has merit but does not fully meet PLOS Global Public Health’s publication criteria as it currently stands. In particular, please note the comments from both reviewers about a through edit of the manuscript for grammatical errors. Therefore, we invite you to submit a revised version of the manuscript that addresses the points raised during the review process.

We look forward to receiving your revised manuscript.

Kind regards,

Sanghyuk S Shin

Academic Editor

Journal Requirements:

Additional Editor Comments (if provided):

Reviewers' comments:

Reviewer's Responses to Questions

**Comments to the Author**

1. If the authors have adequately addressed your comments raised in a previous round of review and you feel that this manuscript is now acceptable for publication, you may indicate that here to bypass the “Comments to the Author” section, enter your conflict of interest statement in the “Confidential to Editor” section, and submit your "Accept" recommendation.

Reviewer #1: All comments have been addressed

Reviewer #2: All comments have been addressed

2. Does this manuscript meet PLOS Global Public Health’s publication criteria? Is the manuscript technically sound, and do the data support the conclusions? The manuscript must describe methodologically and ethically rigorous research with conclusions that are appropriately drawn based on the data presented.

Reviewer #1: Yes

Reviewer #2: Yes

3. Has the statistical analysis been performed appropriately and rigorously?

Reviewer #1: Yes

Reviewer #2: Yes

4. Have the authors made all data underlying the findings in their manuscript fully available (please refer to the Data Availability Statement at the start of the manuscript PDF file)?

Reviewer #1: Yes

Reviewer #2: No

5. Is the manuscript presented in an intelligible fashion and written in standard English?

Reviewer #1: No

Reviewer #2: Yes

6. Review Comments to the Author

Reviewer #1: The data is sound and meets the publication criteria except for the need for a thorough review of grammar, editing and in some places, sentence structure revision.

Overall, the paper needs a major editorial overhaul to correct improper grammar, run-on sentences, poor sentence structure, etc.

Reviewer #2: Congratulations to the authors for a good job.

The authors have sufficiently addressed the comments. The manuscript is rigorous in terms of methodology and ethics. The study's findings were appropriately used to draw conclusions. The issue of data availability has been addressed. Yes, despite the fact that the paper has improved significantly since the earlier comments, grammatical errors exist and must be corrected. In addition, there is an ethical issue that needs to be addressed i.e., how the authors treated anonymity and confidentiality in the study.

7. PLOS authors have the option to publish the peer review history of their article (what does this mean?). If published, this will include your full peer review and any attached files.

**Do you want your identity to be public for this peer review?** For information about this choice, including consent withdrawal, please see our Privacy Policy.

Reviewer #1: No

Reviewer #2: No

---

## [Editor Report · Decision Letter 2]

20 Jun 2024

Perceptions and attitudes towards unmanned aerial vehicles (drones) use for delivery of HIV medication among fisher folk communities on the Islands of Kalangala, Uganda

PGPH-D-23-01417R2

Dear Dr Parkes-Ratanshi,

We are pleased to inform you that your manuscript 'Perceptions and attitudes towards unmanned aerial vehicles (drones) use for delivery of HIV medication among fisher folk communities on the Islands of Kalangala, Uganda' has been provisionally accepted for publication in PLOS Global Public Health.

Thank you for being responsive to the reviewers' comments. I believe they have been adequately addressed.

Best regards,

Sanghyuk S Shin

Academic Editor
